# IL-38: A New Player in Inflammatory Autoimmune Disorders

**DOI:** 10.3390/biom9080345

**Published:** 2019-08-05

**Authors:** Lihui Xie, Zhaohao Huang, He Li, Xiuxing Liu, Song Guo Zheng, Wenru Su

**Affiliations:** 1State Key Laboratory of Ophthalmology, Zhongshan Ophthalmic Center, Sun Yat-sen University, Guangzhou, Guangdong 510000, China; 2Department of Internal Medicine, Ohio State University College of Medicine, Columbus, OH 43210, USA

**Keywords:** interleukin-38, inflammatory autoimmune disease, interleukin-36 receptor, Th17, Treg

## Abstract

Interleukin (IL)-38, a newly discovered IL-1 family cytokine, is expressed in several tissues and secreted by various cells. IL-38 has recently been reported to exert an anti-inflammatory function by binding to several receptors, including interleukin-36 receptor (IL-36R), interleukin-1 receptor accessory protein-like 1 (IL-1RAPL1), and interleukin-1 receptor 1 (IL-1R1) to block binding with other pro-inflammatory cytokines and inhibit subsequent signaling pathways; thereby regulating the differentiation and function of T cells, peripheral blood mononuclear cells, macrophages, and dendritic cells. Inflammatory autoimmune diseases, which are common immune-mediated inflammatory syndromes, are characterized by an imbalance between T helper cells (Ths), especially Th1s and Th17s, and regulatory T cells (Tregs). Recent findings have shown that abnormal expression of IL-38 in inflammatory autoimmune diseases, such as rheumatoid arthritis, psoriatic arthritis, systemic lupus erythematosus, primary Sjogren’s syndrome, psoriasis, inflammatory bowel disease, hidradenitis suppurativa, ankylosing spondylitis, and glaucoma, involves Th1s, Th17s, and Tregs. In this review, the expression, regulation, and biological function of IL-38 are discussed, as are the roles of IL-38 in various inflammatory autoimmune disorders. Current data support that the IL-38/IL-36R and/or IL-38/IL-1RAPL1 axis primarily play an anti-inflammatory role in the development and resolution of inflammatory autoimmune diseases and indicate a possible therapeutic benefit of IL-38 in these diseases.

## 1. Introduction

Interleukin-1 family (IL-1F) members comprise a total of 11 pro-inflammatory cytokines, including IL-1α, IL-1β, IL-33, IL-18, IL-36α, IL-36β, and IL-36γ, and anti-inflammatory cytokines, including IL-1 receptor antagonist (IL-1Ra), IL-36Ra, IL-37, and IL-38 [1]. The dynamic balance of anti- and pro-inflammatory factors determines the development and outcome of inflammation. The imbalance between these factors directly initiates/develops the inflammation or indirectly influences the T helper/regulatory (Th/Treg) cell ratio to eventually affect the progress of inflammation to heighten inflammation [2,3,4,5,6,7]. The recently described cytokine IL-38 was discovered by two research groups in 2001 through the use of a unique high-throughput cDNA sequence scanning approach [8,9]. IL-38 was initially named IL-1HY2 because it is analogous to IL-36Ra (or named IL-1HY1) and IL-1F10, as it was the tenth member of IL-1F, it was renamed to IL-38 in 2010 [8,9,10]. IL-1F includes three subfamilies, and IL-38 is regarded as a member of the IL-36 subfamily (Table 1) together with IL-36α, IL-36β, IL-36γ, and IL-36Ra, based on size, N-terminal structure, and biological function [11]. The IL-38 gene is located at chromosome 2q13-14.1 in the IL-1F cluster between two antagonist genes, the IL-1Ra gene (IL-1RN) and the IL-36Ra gene (IL-36RN) [12]. The IL-38 gene contains five exons, the primary product of which is a 152-amino acid (AA) precursor protein, which was obtained from a fetal skin library, the protein displays a 12-β-stranded trefoil structure. Despite the lack of a signal peptide, which is a trait shared with other IL-1F proteins (IL-1Ra is the only IL-1F member with a signal peptide), IL-38 can be secreted by various cells, such as peripheral blood mononuclear cells (PBMCs), fibroblast-like synoviocytes (FLSs), B cells, keratinocytes (KCs), and various immune cells. Furthermore, IL-38 was demonstrated to be highly expressed in the skin, tonsils, lungs, spleen, heart, placenta, fetal liver, and thymus but at a lower level in inactive immune tissues [9,13]. Mora and colleagues found that IL-38 is secreted in necrotic, apoptotic or inflammatory environments and exerts anti-inflammatory properties, especially on macrophages (Mφs), to inhibit Th17 maturation [14]. In addition, IL-38 exhibits 43% AA sequence homology with IL-36Ra, 37–41% with IL-1Ra [8,9], and 14–30% with IL-1β, as well as other IL-1F members, the predicted three-dimensional structure is analogous to that of IL-1Ra and is thus thought to be a typical antagonist of the IL-1 family along with IL-1Ra, IL-36Ra, and IL-37 (Table 1) [8]. Three candidate receptors, IL-36R, IL-1R1, and IL-1 receptor accessory protein-like1 (IL-1RAPL1) have been proposed for IL-38. IL-38 binds to these receptors, prevents the binding of agonistic ligands and inhibits subsequent signaling pathways, thus exerting anti-inflammatory effects through PBMCs, a human leukemia monocytic cell line (THP-1s), Mφs, and dendritic cells (DCs) to inhibit the activation and function of Th1s and Th17s and promote Treg expansion [14,15,16]. However, the primary pathway is still under dispute.

Inflammatory autoimmune diseases and self-reactive, pathological immune disorders involving the attack of autologous organs, tissues, and cells [17,18] are very common, and both heredity factors and the environment have vital roles in their pathogenic mechanisms [19,20]. Rheumatoid arthritis (RA), systemic lupus erythematosus (SLE), multiple sclerosis (MS), chronic lymphocytic thyroiditis, systemic scleroderma (SSc), primary Sjogren’s syndrome (pSS), psoriasis, and inflammatory bowel disease (IBD) are among the most common inflammatory autoimmune diseases. These diseases, which are a public health problem, are chronic and incurable, leading to both significant individual suffering and a societal burden [19]. Two newly characterized T cell subsets with opposing activities, Th17s and Tregs, play vital roles in immune responses [21,22]. Th17s have pathological relevance in inflammatory autoimmune diseases, whereas Tregs suppress inflammatory responses and can be converted to Th17s, resulting in a high Th17/Treg ratio and inducing autoimmune inflammation [23,24,25]. Corresponding to these observations, increasing Th17 numbers with decreasing Treg numbers have been detected in several common inflammatory autoimmune diseases [26,27]—in contrast to natural T cell precursors, naive T cells, which are defined as a group of T cells not exposed to antigens. Th17s are able to secrete a diverse array of cytokines, such as IL-17A, IL-6, and IL-22, and their differentiation and development are regulated by an independent mechanism [28]. Increasing evidence indicates that IL-38 is a vital element in the pathogenesis of several inflammatory autoimmune diseases, particularly Th17-dependent inflammatory autoimmune diseases [29,30]. Furthermore, IL-38 gene polymorphisms have been reported to be associated with several inflammatory autoimmune diseases, including RA, SLE, psoriatic arthritis (PsA), non-ankylosing spondylitis (AS) and systemic juvenile arthritis ankylosing spondylitis [30,31,32,33,34,35]. A genome-wide association study (GWAS) meta-analysis found that IL-38 correlated strongly with C-reactive protein (CRP) levels [36,37]. Another GWAS reported that IL-38 polymorphisms are related to high IL-1Ra concentrations. In this review, we summarize the current knowledge concerning the roles of IL-38 in several inflammatory autoimmune diseases and also discuss the therapeutic potential of targeting IL-38.

## 2. Functions of IL-38 and Presumed Signaling

The molecular weight of IL-38 is 16.9 kDa, as analyzed by the ProtParam tool [38]. In addition, the self-optimized prediction method from alignment (SOPMA) indicates that IL-38 has a half-life of 7 h, an isoelectric point (pI) of 4.94, and the molecular formula C_757_H_1164_N_198_O_226_S_9_ [39]. Although lacking N- and O-linked glycosylation sites, N-terminal cleavage of IL-38 is required for full activation, similar to most members of IL-1F (Figure 1***)* [9,14]**. However, in contrast to most IL-1F cytokines, IL-38 cannot be cleaved by caspase 1. Further studies should be conducted to identify the protease(s) for IL-38. In its primary soluble form, IL-38 is thought to be a secreted ligand and is mainly observed in two forms: A full-length form called IL-38 (aa 1-152) and a truncated form called IL-38 (aa 20-152) (Figure 1) [14]. In addition, researchers [9] produced recombinant IL-38 (rIL-38) (aa 3-152). Lin and colleagues discovered that IL-38 was produced in a primary form of approximately 25 kDa, but another 17 kDa form exists in mammalian Chinese hamster ovary cells. The form(s) of IL-38 that are naturally present in vivo is unknown [8,9]. Compared with the full-length form, the truncated form of IL-38 displays an increased biofunctionality with inhibited IL-6 production in Mφs [14]. However, Boutet and colleagues [40] found that when full-length IL-38 was overexpressed in THP-1 Mφs or human embryonic kidney (HEK) 293T cells, a large amount of biofunctional IL-38 with anti-inflammatory properties was released. Overall, the specific biofunctions of IL-38 remain unclear. As IL-38 shares considerable sequence homology with IL-1Ra and IL-36Ra, it probably acts as an antagonist. In 2010, Dinarello and colleagues [10] reported that IL-38 inhibits fungal-induced Th17 responses. In 2012, van de Veerdonk and colleagues [16] discovered that IL-17A and IL-22 production was reduced by 37 and 39%, respectively, in PBMC cultures in the presence of IL-38. These reductions were similar to those caused by IL-1Ra (82 and 71%, respectively), as well as those caused by IL-36Ra (23 and 32%, respectively). Moreover, IL-8 (a chemokine that attracts neutrophils and T cells to inflamed tissue) production stimulated by IL-36γ was reduced by 42% in PBMCs in the presence of IL-38, and this effect was similar to that caused by IL-36Ra, which produced a 73% reduction [16]. In 2015, Rudloff and colleagues [30] silenced IL-38 in PBMCs with an IL-38 siRNA, and this treatment led to the increased production of IL-6, APRIL (a proliferation-inducing ligand) and CCL2 in response to Toll-like receptors (TLR) ligands. In 2016, Javier Mora and colleagues [14] found that the IL-38 released into apoptotic cell-conditioned medium (ACM), a type of medium with a large number of apoptotic cells induced by 0.5 µg/mL stauroporine or 0.05 ng/mL anti-Fas [41], decreased the number of inflammatory Mφ-dependent Th17s. In addition, Yuan and colleagues [15] observed that rIL-38 can suppress the production of proinflammatory cytokines, including tumor necrosis factor (TNF)-α, IL-1β, and IL-17, in THP-1 cells.

Nevertheless, several studies have shown that IL-38 can promote the production of proinflammatory cytokines. In 2012, van de Veerdonk and colleagues [16] reported that the suppression of Candida-induced IL-22 and IL-17 expression in human PBMCs induced by IL-38, which produced a dose-dependent response, differed from that induced by classic IL-1Ra (anakinra). Specifically, IL-22 and IL-17 production was inhibited at low IL-38 concentrations but increased modestly at higher concentrations. This may be due to the dimerization of IL-38, which results in a loss of activity. This feature of IL-38 is analogous to that of IL-37, which interacts with IL-18R to recruit a single immunoglobulin IL-1R-related molecule (SIGIRR, an anti-inflammatory factor) to exert an anti-inflammatory effect when present at low concentrations but exhibits contrasting functions at high concentrations [42]. Regardless, Boutet and colleagues observed that IL-38 has anti-inflammatory effects only at high concentrations in vitro, although the specific concentrations used in this study are not publicly available [43]. In 2018, Palomo and colleagues [44] observed that knocking out IL-38 had no effect on the progression or resolution in a model of imiquimod (IMQ)-induced psoriasis and did not cause changes in proinflammatory cytokines, this effect was demonstrated at the cellular level. Overall, the environmental context appears to be critical for IL-38. For example, Mora and colleagues [14] found that when Mφs are cultured in ACM the production of IL-6 and IL-8 is decreased after the addition of truncated rIL-38. When full-length IL-38 is added to apoptotic cell cultures, the full-length cytokine has an anti-inflammatory effect that reduces the production of either IL-6 or IL-8, indicating that in contrast to IL-1, which undergoes procytokine processing under necrotic conditions, rIL-38 is processed under apoptotic conditions [14]. However, Mora and colleagues did not identify the precise cleavage site or determine the specific protease(s), both of which require further investigation.

In summary, IL-38 can have proinflammatory or anti-inflammatory functions depending on its concentration, form(s) expressed, post-translational processing, and local environmental context. When present at low levels, the IL-38 protein exerts anti-inflammatory functions by binding to IL-36R or IL-1R1 to prevent the recruitment of the coreceptor interleukin-1 receptor accessory protein (IL-1RAcP) and/or likely recruits an inhibitory receptor, preventing recruitment of the myeloid differentiation primary response 88 (MyD88) adaptor protein and thus blocking nuclear factor kappa B (NF-κB) or mitogen-activated protein kinase (MAPK) signaling (Figure 1) [16]. Additionally, IL-38 activates an inhibitory coreceptor of the IL-1R family named interleukin-1 receptor accessory protein-like 1 (IL-1RAPL1) to block c-Jun N-terminal kinase (JNK/AP1) signaling-dependent antagonistic effects and then exerts a proinflammatory function by transducing the signal of the inhibitory coreceptor at high levels [14]. The true nature of IL-38 remains controversial, and the coreceptor has yet to be identified. More experimental data are needed to explore the specific effects and functional signaling pathway of IL-38.

### 2.1. The IL-38/IL-1R1 Signaling Pathway and Its Functions

IL-1R1, which is expressed in most cells, is activated by IL-1 procytokines (IL-1α and IL-1β) and blocked by an IL-1R antagonist (IL-1Ra). IL-1R1 is a member of the IL-1R family, which includes IL-1R1 to IL-1R10. IL-1 procytokine interaction with IL-1R1 leads to recruitment of IL-1RAcP and then activation of the intracellular TLR domain. The receptor heterodimer functions as a docking site that induces the MyD88 adapter protein, which ultimately activates the extracellular regulated protein kinases (ERK)1/2, p38 MAPK, NF-κB and JNK signaling pathways, thereby triggering the secretion of inflammatory cytokines (Figure 1) [45,46]. In contrast, IL-1Ra prevents IL-1R signaling by both blocking IL-1RAcP recruitment and employing SIGIRR, interfering with downstream signaling (Figure 1). The IL-38 protein shares 41% AA homology with naturally occurring IL-1Ra, which indicates that IL-38 probably has anti-inflammatory properties similar to those of IL-1Ra [8]. In 2001, Bensen and colleagues [8] demonstrated that IL-38 is able to bind to IL-1R1, nevertheless, the binding affinity of IL-38 is much lower than that of IL-1 procytokines and IL-1Ra. In 2012, van de Veerdonk and colleagues [16] observed that IL-38 cannot bind to immobilized IL-1R1. In 2016, Mora and colleagues [14] found that truncated IL-38 exhibits relatively strong affinity for IL-1R1 and that truncated IL-38 decreases IL-6 release by Mφs induced by IL-1β but full-length IL-38 does not demonstrate the same ability. Adding the supernatant from IL-38-overexpressing THP-1 cells reduces IL-6 production by synovial fibroblasts (SFs) stimulated with IL-1β [40]. In conclusion, the IL-38/IL-1R1 pathway likely exerts anti-inflammatory effects, although none of these reports have proven that IL-1R1 is essential for IL-38-related bioeffects. Thus, the role of IL-1R1 remains to be further studied with regard to the broad expression of IL-1R1 in immune cells.

### 2.2. The IL-38/IL-36R Signaling Pathway and Its Functions

IL-36R is the specific receptor of IL-38. However, although IL-38 can bind to IL-36R and act as a receptor antagonist, this property is weak and has a dose-dependent feature, thus, IL-38 can be thought of as a partial antagonist of IL-36R. IL-36R, which is activated by IL-36 procytokines (IL-36α, IL-36β and IL-36γ) and inhibited by IL-36Ra, is also named IL-1R6 or IL-1Rrp2 and is mainly expressed by DCs, SFs, KCs and Ths [29,47,48]. For both IL-36Ra and IL-36 procytokines, nine AAs need to be cleaved before the A-x-Asp motif becomes fully active [49]. IL-36 procytokines interact with IL-36R and recruit IL-1RAcP, which forms a cytoplasmic toll-interleukin 1 receptor (TIR) domain [11]. The TIR domains function as a docking site for MyD88 [50], which activates a series of chain reactions that ultimately result in the activation of several transcription factors, an outcome similar to that of the IL-1R1 pathway (Figure 1). This process triggers TNF-α, IFN-γ, and IL-17, among others, to promote neutrophil influx, DC excitation, Th1 and Th17 polarization, and KC proliferation [51]. Moreover, IL-36 procytokines elevate IL-23 production to promote Th17 expansion [52] and are regarded as a part of the IL-17 framework because they are not only regulated by IL-17 but also enhanced by IL-17 functions [51]. In contrast, IL-36Ra displays an antagonistic effect on several inflammatory autoimmune diseases [29]. IL-36Ra blocks signaling downstream of IL-36R by preventing recruitment of IL-1RAcP, and at the same time, IL-36Ra recruits SIGIRR, similar to IL-1Ra (Figure 1) [29,53]. As with IL-36Ra, IL-38 blocks downstream signaling by preventing binding of IL-36 procytokines to IL-36R and recruitment of IL-1RAcP, but whether IL-38 has the ability to recruit SIGIRR has not yet been reported. However, it appears very likely that IL-38 has a dose-dependent effect similar to that of IL-37 that can recruit SIGIRR for an anti-inflammation effect, which requires further research for validation. In addition, whether IL-38 acts as a pro-inflammatory or anti-inflammatory cytokine in the IL-38/IL36R pathway still needs to be determined. In 2012, van de Veerdonk and colleagues [16] discovered that IL-38 can bind only to IL-36R but not IL-1RI, IL-18R and IL-1RAcP, the ability of IL-38 to bind with IL-36R is also similar to that of IL-36Ra. Administration of rIL-38 inhibits heat-killed *Candida albicans*-induced Th17 responses with decreasing IL-6, IL-8, and IL-17 levels in PBMC cultures, and these outcomes are similar to those caused by IL-36Ra. Higher IL-6 and IL-8 concentrations were observed after knocking down the IL-38 gene [16]. Additionally, the IL-38/IL-36R axis has been reported to have a vital role in the pathogenic mechanism of many diseases, including inflammatory autoimmune diseases such as SLE [54], RA [29], AS [55], IBD [29], psoriasis [56], pSS [57], asthma [58], retinopathy [59], hepatitis [60], coronary artery diseases and cancer [61], and myocardial infarction [62]. IL-38 also induces proinflammatory cytokines in response to lipopolysaccharide (LPS) in DCs [14,16]. Moreover, Palomo and colleagues [44] observed that IL-38/IL-36R appears to have no effect on a mouse model of skin immunity. In addition, IL-36R does not appear to be relevant in several mouse models of arthritis [63,64], although IL-38 can reduce clinical scores and IL-17 expression in these models [40]. More studies should be performed to resolve these discrepancies, and future studies should pay more attention to DCs, as well as lymphocytes, because of their essential roles in the IL-38/IL-36 axis [43,65].

### 2.3. The IL-38/IL-1RAPL1 Signaling Pathway and Its Functions

Using receptor-binding assays, Mora and colleagues [14] found that IL-38 binds to IL-36R, as well as IL1RAPL1, which is highly relevant to neurobiology, immunity, and tumor biology and is implicated in cerebellar development, as well as in cognitive defects [66,67]. Although IL1RAPL1, termed TIGIRR-2 or IL-1R10, is an orphan receptor that differs from other IL-1R family members by having three extracellular Ig domains and an intracellular TIR domain, it does belong to the IL-1R family along with SIGIRR and TIGGRR-1 (IL-1R9) [45]. In contrast to other IL-1R family members, which usually activate MAPK or NF-κB signaling, IL1RAPL1 activates the JNK/AP1 pathway [68]. Both the full-length and truncated forms of IL-38 bind to IL1RAPL1. However, full-length IL-38 binds to IL1RAPL1, activating the downstream JNK/AP1 pathway and then increasing IL-6 production to exert a stimulatory effect, conversely, truncated IL-38 diminishes JNK/AP1 signaling and limits Th17 activation to reduce inflammatory Mφ activation by decreasing IL-6 and IL-8 levels (Figure 1) [14]. The mechanism by which IL1RAPL1 induces AP1 activation is unknown. It may elevate the basal activity of factors or produce an endogenous coreceptor of IL1RAPL1 such as IL-1RAcP, IL-1R1 or IL-36R to activate AP1, which explains why IL-38 can interact with several receptors. More studies should be performed to elucidate this mechanism. In addition, apoptotic cells can produce high levels of IL-38 protein, which binds to IL1RAPL1 to decrease AP1 activation in Mφs by antagonizing JNK phosphorylation upon interaction with apoptotic cells and reduce IL-6 and IL-8 concentrations.

## 3. The Role of IL-38 in Inflammatory Autoimmune Diseases

### 3.1. The Role of IL-38 in RA

RA is recognized as a chronic inflammatory autoimmune disease with synovial inflammation as well as progressive cartilaginous and joint architecture damage [69]. Joint malfunction, cartilage damage and debilitating pain are typical symptoms of RA patients [70]. The predominance of Th1s and Th17s over Tregs plays a pathogenic role in RA [71,72], and one study reported that the frequencies of Th17s are closely related to clinical scores in RA patients [73]. In RA, large Th populations permeate the synovial membrane, where they produce vast amounts of IFN-γ, IL-1β, IL-6, and IL-17, among others. These cytokines are thought to be crucial for driving inflammation and the destructive process of RA [70,74]. Thus, inhibiting these cytokines can reduce RA manifestations and slow the progression of RA. In contrast, IL-4 and IL-10 have been proposed to improve arthritis [70]. IL-38 expression is highly increased in the serum and synovial membrane of RA patients compared with normal subjects [75,76,77]. Moreover, IL-38 concentrations are elevated in the synovial fluid (SF) [29,75]. Notably, synovial levels of IL-38 are higher than the serum levels. IL-38 is strongly expressed in the synovial lining [76], and elevated IL-38 expression is associated with TNF-α, IL-1β, IL-6 [76], IL-1Ra, CCL3, CCL4, M-CSF [29], erythrocyte sedimentation rate (ESR), and CRP, and probably serve as a potential biomarker for RA [78]. Moreover, expression of IL-38 is reduced after treatment [78], and IL-38, IL-36 procytokines and IL-36Ra are correlated with the severity of RA [29,78]. IL-36 procytokines, IL-36Ra and IL-36R levels are highly increased in the synovial membrane of RA patients [29,77], and IL-36R is constitutively expressed in human SFs and articular chondrocytes [79]. IL36α/β induces IL-6 and IL-8 production in SFs through MAPK/NF-κB activation [79,80], and IL-36 procytokines, IL-36Ra and IL-38 are related to each other [29]. Taken together, these findings reveal a probable characteristic of the IL-38/IL-36R pathway in RA. Moreover, IL-1β can increase the severity of joint destruction in RA [31], and IL-1 and IL-6 indirectly promote inflammation by downregulating the stability and functionality of Tregs [5,81]. rIL-1Ra can be applied for the treatment of systemic juvenile idiopathic arthritis and RA [16]. Moreover, IL-1Ra knockout mice spontaneously develop chronic inflammatory arthropathy [82], which indicates that IL-1R1 plays a role in RA and that the IL-38/IL-1R1 axis may be involved. One research group noted that in Western blot analyses, the IL-38 protein acquired from the synovial membrane of RA patients is 14–15 kDa [29], which contrasts with the results of others that functional rIL-38 is 17–18 kDa [40,76]. Overall, the relationship between high levels of IL-38 and RA indicates a potential role of IL-38 in the progression or development of the disease. In addition, IL-38 gene polymorphisms influence RA susceptibility, and Korean patients carrying the IL-38 single-nucleotide polymorphism (SNP) rs3811058 are susceptible to developing RA [31].

To illuminate the effect of IL-38 on RA, many studies have been conducted using both mouse models and in vitro experiments. Recently, several types of RA experimental murine models, such as collagen-induced arthritis (CIA), collagen antibody-induced arthritis (CAIA), antigen-induced arthritis (AIA), and K/BxN-serum-transfer-induced arthritis (STIA), have been used. In the CIA mouse model, IL-36 procytokines, IL-36Ra and IL-38 levels are elevated [29]. Clinical scores for inflammation are significantly reduced with decreasing Th17 cytokine, chemokine ligand (CXCL) 1, receptor activator of nuclear factor kappa-B ligand (RANKL) (osteoclastic cytokine), and TNF-α levels after articular injection of an adeno-associated virus (AAV) carrying IL-38 [40]. Additionally, inhibiting Th17 expansion by neutralizing IL-6 suppresses the development of CIA [83]. Interestingly, IL-36 procytokines and IL-36Ra mRNA concentrations are strongly increased in CIA mice joints at the peak of inflammation, whereas IL-38 expression is induced primarily at the time of inflammation resolution, which might be important for reducing joint swelling and Mφ infiltration [76]. However, in contrast with IL-1Ra expression, which has an obvious impact on bone erosion in mice [84,85], overexpression of IL-38 has no influence on bone damage [40]. Similarly, blocking IL-1 procytokines protects bone and cartilage from destructive processes in RA patients [85]. The lack of an effect on cartilage or bone damage by IL-38 might be a result of the decrease in IL-10 [40], which is an anti-inflammatory and anti-osteoclastogenic cytokine that prevents bone damage. The use of an anti-IL-1R1 antibody suppresses the progression as well as the severity of CIA [64]. In addition, mice with IL-1Ra conditionally knocked out in the myeloid compartment exhibit exacerbated CIA development with increased IL-17 levels in their joints [86]. In the STIA mouse model, clinical scores for inflammation were significantly reduced after injecting AAV encoding IL-38 [40]. Moreover, IL-38 knockout mice display more severe symptoms of STIA with enhanced joint inflammation, as well as higher IL-1β and IL-6 concentrations, than those of wild-type mice [76]. The pathogenic mechanism of STIA is not related to B cells or T cells, which is in accord with the fact that IL-38 affects innate immune cells such as Mφs [87]. In the CAIA model, IL-36 procytokines and IL-36Ra mRNA levels are increased in a manner similar to those in the CIA model [40]. In the AIA mouse model, blocking IL-17 in the initial phase of AIA inhibits AIA development [88]. However, articular injections of IL-38 have no obvious effect on knee diameter or clinical score [40]. In addition, it appears that the ability of IL-1Ra to decrease clinical scores is less efficient in AIA than in CIA [89]. These results demonstrate that different mouse models of RA have different orientations toward IL-1F members. Both IL38/IL-1R1 and IL-38/IL-36R play roles in the pathogenic mechanism of RA, yet Lamacchia and colleagues reported that the injection of an anti-IL-36R antibody had no significant effect on RA. Similar results were observed in IL-36R knockout mice, and the researchers failed to find associations between IL-36R, IL-36Ra and IL-36γ expression levels and the severity of arthritis in K/BxN STIA, AIA and CIA models [64]. Moreover, Derer and colleagues [63] reported that although IL-36α and IL-36R levels are increased in the joints of TNF-induced RA mice, there is no change in histology, disease or bone homeostasis after blocking IL-36 signaling. The possible reason may be that the blockade of IL-36R suppresses not only IL-36 procytokines but also their antagonists, IL-36Ra and IL-38, and, therefore, has a neutralizing role.

In vitro, IL-38 and TLR4 levels were found to be significantly increased in PBMCs. Moreover, IL-38 can suppress LPS-mediated expression of TLR4, IL-6, IL-8 and TNF-α in RAW264.7 cells [75]. IL-6, IL-23, TNF-α and IL-10 expression is significantly reduced after LPS stimulation in THP-1 Mφs overexpressing human IL-38 [40]. Additionally, conditioned medium collected from THP-1 cells overexpressing IL-38 or from epithelial HEK 293T cells overexpressing IL-38 can significantly decrease IL-6, TNF-α and IL-23 production in Mφs and SFs from RA patients [40]. IL-36 cytokines are expressed by monocytes and inflammatory Mφs. IL-36 procytokines induce IL-6 and IL-8 production in FLSs [29,79]. Overall, the IL-38/IL-36R axis performs its anti-inflammatory function by suppressing proinflammatory cytokines from Mφs, PBMCs or FLSs, and RA-SFs induced by IL-1β significantly induce inflammation, indicating that IL-38/IL-1R1 may be involved in this process.

Together, the current evidence clearly shows that increased IL-38 expression potentially limits RA severity, which emphasizes the necessity of enhancing IL-38 as a therapeutic target in RA. The IL-38/IL-36R or IL38/IL1R1 pathway may be involved, but more information is necessary to validate the effect of these pathways. IL-38 may predominantly impact PBMCs or Mφ infiltration and the secretion of key cytokines and chemokines implicated in the Th17 pathway by these cells and thus, exert anti-inflammatory effects on RA (Figure 2).

### 3.2. The Role of IL-38 in Psoriasis

Psoriasis is a prevalent autoimmune disorder of the skin featuring red, scaly lesions or plaques with acanthosis, epidermal KC hyperplasia, parakeratosis, high-level inflammatory infiltration, and increased cytokine and chemokine expression [90]. Although the pathogenic mechanism of psoriasis remains unclear with regard to many aspects, it is considered to be a T-cell-mediated skin inflammation disease, and Ths are crucial for the development of psoriatic lesions [91]. Further studies have demonstrated that Th1 and Th17 cytokines play pathogenic roles in psoriasis [51,92]. PsA, which is characterized by erosion, as well as aberrant bone apposition, and is closely linked to psoriasis in the skin, is a heterogeneous joint disorder. The IL-23/IL-17 pathway exerts critical pathogenic effects on both psoriasis and PsA [93]. One study reported that the frequencies of Th17s are closely related to clinical scores in psoriasis and PsA [73]. In contrast, Tregs decrease expression of Ths, ameliorating psoriasis hyperproliferation [94]. The imbalance between Th17s and Tregs leads to psoriasis. IL-38 expression is highly expressed in skin and is evidently reduced both in the skin and peripheral blood of psoriatic patients [29,95,96]. Furthermore, IL-38 is produced by PBMCs and related to disease severity [97]. Li and colleagues found that patients with pustular psoriasis (a severe form of psoriasis) exhibit high serum levels of IL-38 [56]. In addition, SNPs in the region encoding IL-38 are linked to a higher susceptibility to PsA [33]. Nonetheless, Keermann and colleagues [98] demonstrated that compared with control skin, psoriatic skin did not exhibit a change in IL-38 expression. Although the function of IL-38 is disputed, studies have shown that other IL-36 family members are vital in psoriasis [90]. IL-36 procytokine, IL-36R and IL-36Ra levels are highly increased in inflamed skin, especially in human psoriatic lesional skin [92,99], and correlate with disease severity [29]. Furthermore, psoriatic lesional skin improves after inhibiting IL-36R [100], and treating psoriasis patients with etanercept (an anti-TNF-α drug) leads to clinical improvement in accordance with significantly decreased IL-36 procytokine, IL-36R and IL-36Ra mRNA levels [92]. In addition, IL-36Ra-deficient patients generally develop a severe form of psoriasis known as pustular psoriasis [101]. Taken together, (although one study found no change in IL-38 expression between psoriatic patients and healthy controls), most of the observed connections between IL-38 production and disease activity indicate that the IL-38/IL-36 axis may participate in psoriatic development, but more studies are needed to confirm this conclusion.

IL-38 is highly expressed in mouse skin [95], and IL-36 procytokines and IL-36Ra mRNA levels are increased in various mouse models of psoriasis [92], with elevated levels of IL-1β, IFN-γ, IL-17, and IL-22, among others [51]. Furthermore, IL-36 procytokines and IL-36Ra levels are further increased at the peak of psoriasis in a mouse model induced by IMQ, a TLR7 agonist that activates the innate immune response and induces skin inflammation, although IL-38 expression is reduced dramatically at the same time, these findings are in accordance with those observed in psoriasis patients [29]. Palomo and colleagues [44] recently reported that knocking out IL-38 in mice has no effect on the progression or resolution of IMQ-induced inflammation, which may be due to the few T cells present in ear and tail skin and that psoriasis in humans rarely involves the ears [95]. Han and colleagues [95] reported that knocking out IL-38 in mice results in delayed disease resolution with exacerbated IL-17-mediated inflammation and that this situation can be reversed after administrating human rIL-38 or γδ T cell-receptor-blocking antibodies. These authors also found that IL-1RAPL1 is upregulated upon γδ T cell (another type of IL-17 producing T cell in addition to Th17 cells) activation and is required for IL-38 to suppress γδ T cell IL-17 production. In addition, IL1RAPL1 knockout mice show reduced inflammation and IL-17 production by γδ T cells. IL-36Ra knockout mice develop more severe skin pathology than wild-type mice do [101]. In a Th17-mediated psoriasis mouse model, an anti-IL-22 antibody attenuated symptoms and reduced expression of IL-36, whereas injection of rIL-22 had the opposite effects [51], suggesting that IL-22 can synergize with the proinflammatory effects induced by IL-36 procytokines. Although IL1Ra-deficient BALB/c mice have features in common with psoriasis [102], mice deficient in IL1R1 and IL-1RAcP have no obvious skin abnormalities [103,104], suggesting that the IL38/IL-1R1 axis might not be involved in the pathogenic mechanism of psoriasis.

In vitro, KCs treated with TNF-α, IFN-γ, IL-22 and IL-17A [51] are regarded as the major source of IL-36 procytokines, IL-36Ra and IL-38 [29]. Furthermore, IL-36 procytokines directly induce expression of Th17 cytokines, proinflammatory cytokines and chemokines TNF-α, IL-6, IL-8, CCL17, CCL22 and CXCL8 in KCs [16,105] and increase IL-17A-mediated induction of antibacterial peptides [51]. However, adding recombinant full-length IL-38 can inhibit the above biological processes induced by IL-36 procytokines in KCs [96]. Overall, IL-36 stimulates KCs to regulate skin inflammation, in addition, IL-36 forms a feedback loop with Th17 cytokines that incessantly drives cytokine expression in psoriatic tissue, which can be broken by IL-38. Dedifferentiated KCs triggered by the inflammatory cytokines IL-36γ, IL-17, and IL-22 show decreased IL-38 expression, which is relevant to the loss of CK10, a marker of KC differentiation and expression [29,96,106]. Additionally, Th17 cytokines inhibit KC differentiation [107]. IL-38 might inhibit psoriasis by restoring the differentiation of KCs and reducing the infiltration of immune cells [96]. These data indicate that loss of IL-38 in the epidermis plays a significant role in the pathogenic mechanism of psoriasis.

In summary, IL-36 cytokines play a vital role in the pathogenesis of psoriasis. Moreover, IL-38 may exert an anti-inflammatory effect by inhibiting Th17 responses (Figure 2). The IL-38/IL-36R axis is important in this hyperproliferative skin autoimmune disorder. As the role of IL-38 remains controversial, more relevant cellular and animal studies are necessary to explore the activation and molecular mechanism of IL-38 in psoriasis.

### 3.3. The Role of IL-38 in SLE

SLE, a severe multiorgan inflammatory autoimmune disease with clinical manifestations such as fever, mucosal skin lesions, arthritis, and kidney and neurologic syndromes, is a common chronic inflammatory disease among young females [108]. The imbalance of Th17s/Tregs is vital to the pathogenic mechanism of SLE [109]. The concentrations of IL-38 in SLE patients are higher than those in sera and are strongly related to disease severity [30]. Moreover, in comparison to those with inactive SLE, patients with active SLE have significantly increased IL-38 levels [30], and the concentration of IL-38 is closely correlated with the risk of renal and central nervous system complications [30]. However, recently, Takeuchi and colleagues observed that among 18 juvenile-onset SLE patients, only one exhibited elevated IL-38 levels regardless of lupus nephritis severity, even though the IL-38 levels of that patient decreased after treatment [54]. Moreover, expression of IL-36 procytokines in the plasma is greatly increased in active SLE patients compared with controls and positively related to SLE disease activity as well as increased IL-10 concentrations [110,111].

In studies of the Murphy Roths Large (MRL)/lpr mouse model, IL-38 expression was found to be significantly lower in these mice compared to in control mice [112]. The intravenous injection of rIL-38 attenuated clinical severity [112], and the histopathological characteristics of decreased skin inflammation and nephritis were observed [30]. Th17 numbers, Th17-regulated gene mRNA levels and serum concentrations of CXCL10, IL-1β, IL-6, IL-17, and IL-22 were notably reduced but the mRNA expression of Treg-regulated genes evidently increased, although the expression levels of Th1- and Th2-regulated genes were not different after IL-38 treatment (Figure 2) [112]. Overall, IL-38 treatment ameliorates skin inflammation and nephritis in SLE mice, most likely by inhibiting IL-17 and IL-22. In vitro observations have shown that PBMCs with IL-38 expression knockdown release up to 28-fold more IL-6, CCL2, and APRIL than control cells do upon stimulation of TLR-7 and TLR-9 [30].

Overall, IL-38 plays a vital role in the pathogenic mechanism of SLE and probably functions as a predictive and protective cytokine. IL-38 might function via the IL-38/IL36R pathway, and it may represent a promising target for therapeutic intervention in SLE. More studies are needed to confirm this hypothesis and to explore the underlying molecular mechanisms.

### 3.4. The Role of IL-38 in IBD

IBD, a chronic refractory or relapsing intestinal disease that can manifest as diarrhea, abdominal pain, and even bloody stools, imparts significant morbidity, consisting of ulcerative colitis (UC) and Crohn’s disease (CD), in Western society [113,114]. IL-38 was found to be highly expressed throughout the mucosa, submucosa, muscular, and serosa layers of IBD patients [115]. In addition, IL-38 expression was much higher in active UC compared to active CD [115]. The analysis of colonic biopsy specimens has shown that levels of IL-36α/γ and IL-38 but not IL-36β are increased in active CD patients and related to IL-1β and IL-17A [29]. IL-36 procytokine levels are also enhanced in active lesions in UC patients [50,115]. IL-36Ra expressed by only lamina propria mononuclear cells (LPMCs) is significantly reduced in the colon of UC patients [114], and DNA microarray analysis has identified IL-36γ as the most preferentially expressed cytokine in inflammatory colonic Mφs [116]. These results indicate that IL-36 procytokines, IL-36Ra and IL-38 play a partial role in the pathogenesis of IBD.

Several studies have explored the effect of IL-36 cytokines in disease development using IBD mouse models. In mice with dextran sulfate sodium (DSS)-stimulated colitis, IL-36α/γ and IL-38 levels are increased in the colon, although IL-36β and IL-36Ra levels are not [29,114]. Compared with wild-type mice, IL36R knockout mice display decreased intestinal inflammation, which is related to reduced inflammatory cell infiltration. In addition, it has been reported that compared with wild-type, the IL-36R knockout mouse model of infectious colitis exhibits altered Th responses with increased Th17 and decreased Th1 responses mediated by decreased expression of IL-2, a cytokine that can efficiently stimulate Th1 differentiation and suppress Th17 development [114]. Another report showed that compared with wild-type mice, IL-36R knockout mice display decreased inflammation but defective recovery after DSS-stimulated damage, which was related to neutrophil infiltration in the wound and a reduction in IL-22 expression [116,117]. The administration of an aryl hydrocarbon receptor agonist restores IL-22 expression and promotes recovery in IL-36R-deficient mice after DSS treatment [117]. Another study found that IL-36 cytokines have no significant effects on Th1 and Th2 polarization of LPMCs derived from IBD mucosa [50].

Nishida and colleagues [50] discovered that IL-36α/γ interacts in vitro with IL-36R to induce CXC chemokine expression in the human intestinal epithelial cell line, HT-29, in a dose-dependent and time-dependent manner. In addition, the loss of IL-36R leads to an increase in the Th17 response in the primary phase of infection [114], enhanced Th1 responses and inhibited Th17 responses were observed after adding IL-36α in vitro (Figure 2) [114].

Although most studies have focused on the role of IL-36 procytokines, few have investigated the role of IL-38 in IBD. Overall, current data identify an innovative role of the IL-36 pathway in the regulation of colonic bowel inflammation and post-injury healing. It is likely that IL-38 functions as an anti-inflammatory cytokine in IBD, based on the present evidence, which shows that the IL-38/IL-36R pathway may be a new direction for therapeutic intervention in IBD, further study is needed to confirm that conclusion.

### 3.5. The Role of IL-38 in Other Inflammatory Autoimmune Diseases

As the nature of IL-38 in common inflammatory autoimmune diseases has been well demonstrated, IL-38 has been suggested to be a biomarker for the development of other inflammatory autoimmune diseases. IL-38 and IL-36α concentrations are elevated in the salivary gland of patients with pSS, and IL-36α expression is highly increased in the serum and is related to the activity and tissue inflammation of the disease. Recent studies have demonstrated that the IL-23/IL-17/IL-22 axis is critical for pSS (Figure 2) [118]. Thus, the IL-38/IL-36R pathway may play an important role in pSS [57].

IL-38 expression is significantly increased in perilesional and lesional areas, but IL-36 procytokines and IL-36Ra levels are significantly elevated only in the lesional area in hidradenitis suppurativa (HS), an autoinflammatory neutrophilic pustular skin disease [119]. The IL-23/Th17 pathway is upregulated in HS [120]. In addition, IL-38 appears to exert a protective effect on HS [119]. Thus, the IL-38-IL36R axis likely plays a role and is a potential therapeutic target in HS.

AS is a type of autoimmune arthritis characterized by inflammation in the spinal and sacroiliac joints, which induces serious disability [121]. The IL-38 polymorphism rs3811058 is related to AS in both the Taiwanese Chinese [34] and Chinese Han [35] populations. In 2012, Lea and colleagues found that Europeans, but not Asians, with the IL-38 polymorphism rs3811058 (IL-1F10.3) are susceptible to developing AS [55], which contradicts previous data. In addition, a study discovered that the IL-38 gene is correlated with non-AS in the French population [32].

Glaucoma has recently been considered to represent an immune-medium inflammation disease [122]. IL-38 and IL-36 procytokines are increased in the aqueous humor of chronic primary angle-closure glaucoma and correlate with the mean deviation of the visual field of glaucoma [123].

Recently, Xu and colleagues [118] found that IL-38 can inhibit obesity and improve insulin resistance induced by a high-fat diet.

The above findings display a possible protective effect of IL-38 in inflammatory autoimmune diseases. However, the limited data concerning this field necessitate establishment of better animal models to explore the actual functions and underlying immune mechanisms of IL-38 and its receptor in these contexts.

## 4. Conclusions

### 4.1. IL-38 and Inflammatory Autoimmune Diseases

IL-38, a recently recognized IL-1F member, is extensively expressed by immune cells and plays a crucial role in a diverse array of inflammatory autoimmune diseases, however, its exact signaling and functional pathway remains poorly understood since its discovery sixteen years ago. IL-38 is considered to be an antagonist similar to IL-1Ra and IL-36Ra, which mainly inhibit Th17 cytokines and are associated with disease severity and treatment, suggesting that IL-38 might represent a potential biomarker for predicting inflammatory autoimmune disease development and the clinical efficacy of inflammatory autoimmune disease therapeutics. This review summarizes current studies on the emerging role of IL-38 in various inflammatory autoimmune diseases (Table 2). IL-38 might not function as a specific antagonist but may instead be a broad anti-inflammatory cytokine, as one study found that IL-38 can bind to IL-36R, IL1R1 and IL-1RAPL1 in vitro, in contrast, most in vivo data are related to the IL-36R-mediated pathway. Binding of IL-38 to its receptors blocks the interaction with procytokines to reduce inflammation. Moreover, considering the dose-dependent effect of IL-38, low concentrations of this cytokine may recruit an inhibitory coreceptor, whereas high concentrations recruit a signal transducing coreceptor. Although the role of IL-38 in inflammatory autoimmune diseases is controversial, we can nonetheless identify patterns and form conclusions (Table 2).

### 4.2. IL-38 and Immune Cells

Multiple studies have shown that IL-38 influences several immune cell types involved in autoimmunity. In this review, we discuss way that, in the inflammatory autoimmune environment, IL-38 exerts a protective role by preventing activation and function of Th1s and Th17s, thus inhibiting Th1, Th17, and γδT cell cytokines (TNF-α, IL-1β, IFN-γ, IL-17A) and downregulating IL-6, IL-8, IL-22, and IL-23 expression. IL-17 is able to induce the expression of IL-1β, TNF-α, IL-6, CXCL1, and CCL20, among others [124,125]. Additionally, IL-6 is necessary for the differentiation and expansion of Th17s, and altered IL-6 production in Mφs affects Th17 maturation [126,127]. IL-23, which is produced by Mφs and DCs, induces the polarization and maintenance of Th17s [128]. Tregs are another important regulator of autoimmunity, and IL-38 contributes to the proliferation of Tregs and prevents these cells from transforming into Th17s, thus increasing the number of Tregs. Although IL-38 is reportedly secreted by B cells, its direct effects on B cells need to be investigated under autoimmune conditions. At low concentrations, IL-38 decreases IL-17, IL-22, and IL-36γ-derived IL-8 levels in PBMCs, but at high concentrations, IL-38 increases IL-17 and IL-22 levels. Silencing IL-38 in PBMCs enhances IL-6, APRIL, and CCL2 production. Furthermore, IL-38 elevates IL-6 levels in DCs, and these cells should be studied in more detail in the future because they express high levels of IL-36R and play a key role in mediating Th17 polarization [65]. The IL-38 released into ACM decreases the number of Th17s dependent on inflammatory Mφs. Additionally, IL-38 can suppress TNF-α, IL-1β and IL-17 production in THP-1 macrophages. IL-38 depletion enhances expression of IL-6 and IL-8 in Mφs, and IL-38 can increase KC differentiation and inhibit the proliferation, migration, and tube construction in vascular endothelial growth factor (VEGF)-induced endothelial cells. Overall, IL-38 plays key roles in perpetuating autoimmunity by acting on a variety of cells.

### 4.3. Concentrations/Dose and Forms of IL-38 on Its Biofunction

Under which conditions does IL-38 exert antagonistic effects and proinflammatory effects? In general, the effect of the concentration/dose and forms of IL-38 on its function should be considered. For example, Van de Veerdonk and colleagues [16] found that IL-22 and IL-17 production in PBMC cultures was inhibited at low IL-38 concentrations of 10 and 100 ng/mL but that the degree of IL-22 inhibition was large at 10 ng/mL compared to 100 ng/mL and increased at high concentrations of 1 µg/mL. Additionally, Han and colleagues [95] found that IL-17 induced by C. albicans in human PBMCs was reduced by mature IL-38 (aa 7–152) at a concentration range from 0.1 to 250 ng/mL and that the inhibitory effect gradually decreased at higher concentrations (500 and 1000 ng/mL). These researchers also reported that full-length IL-38 (aa 1–152) exerted anti-inflammatory effects at concentrations of 250 ng/mL. Mora and colleagues [14] also found that IL-6 production in human macrophages was inhibited by truncated IL-38 (aa 20–152) at concentrations ranging from 0 to 20 ng/mL stimulated by IL-1β or LPS in a dose-dependent manner. However, they also found that IL-6 production in human Mφs was significantly increased by full-length IL-38 (aa1–152) ranging from 0 to 20 ng/mL, especially at 20 and 10 ng/mL, stimulated by IL-1β or LPS, full-length IL-38 (10 ng/mL) was also capable of decreasing IL-6 and IL-8 levels when Mφs were exposed to ACM. In another study, Wang and colleagues [129] found that IL-38 suppresses the migration, invasion, and growth of non-small cell lung cancer cells in a dose-dependent manner at a concentration ranging from 0 to 100 ng/mL. Zhang and colleagues [59] reported that IL-38 suppresses VEGF-induced proliferation and migration of endothelial cells at a concentration of 1 or 5 ng/mL, and Yuan and colleagues [15] reported that IL-17, TNF-α, IL-1β, and MCP-1 mRNA levels were inhibited by IL-38 at concentrations of 20 ng/mL in THP-1 cells exposed to LPS. In summary, we conclude that low concentrations of IL-38 (below 250 ng/mL, particularly 1–100 ng/mL) exert an anti-inflammatory effect but that high concentrations (above 250 ng/mL, particularly 1 µg/mL) exert a pro-inflammatory effect, truncated IL-38 (aa 20–152) and mature IL-38 (aa 7–152) also have an anti-inflammatory effect, whereas full-length IL-38 (aa1-152) has a pro-inflammatory effect. Full-length IL-38 can be cleaved to exert an anti-inflammatory effect in an apoptosis environment. Thus, the administration dose and the form of IL-38 should be considered when using IL-38 in the future.

### 4.4. Problems to Be Solved Regarding IL-38

Most studies conclude that IL-38 exerts anti-inflammatory effects in several autoimmune disorders, including RA, psoriasis, SLE, and IBD. Potential mechanisms include IL-38/IL-36R-mediated dysfunction of Th1s/Th17s and promotion of Treg differentiation. However, data from several studies show that IL-38 plays a protective role in inflammatory autoimmune diseases, as well as in other diseases. Overall, numerous controversies and questions remain, such as the following: Which intracellular or extracellular processes are IL-38 involved in, and are these processes significant for displaying the antagonist or agonist function of IL-38? Which proteases are involved in these processes? Is the natural form of IL-38 in vivo the full-length and/or truncated form and/or another form, and what effect does each have? Does IL-38 recruit SIGIRR or other inhibitory receptors after binding to IL-36R, IL-1RAPL1 or IL-1R1? Why is IL-38 expression mainly increased during the resolution of inflammation? These unanswered questions indicate that the immune agent IL-38 is elusive in inflammatory autoimmune diseases. In summary, IL-38 and its receptors IL-1R1, IL-36R, and IL-1RAPL1 are molecules of interest in studies of inflammatory autoimmune diseases. We conclude that the IL-38/IL-36R axis primarily plays an anti-inflammatory role in the development and resolution of inflammatory autoimmune diseases. Nevertheless, it remains unclear whether IL-38 and its receptors may serve as biomarkers that predict disease severity and activity. Roles of IL-38 differ across diverse diseases and/or in different disease phases. Given the current limited data, these problems should be prioritized in future studies on inflammatory autoimmune diseases. We propose additional research to demonstrate the precise function and detailed mechanisms of IL-38 in inflammatory autoimmune diseases.

## Figures and Tables

**Figure 1 biomolecules-09-00345-f001:**
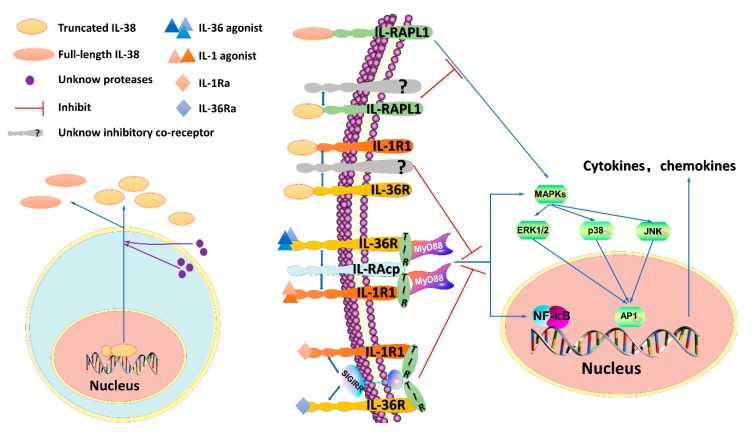
IL-38 signaling pathways. The N-terminus of IL-38 requires cleavage, but which protease(s) is responsible remains unknown. The process may involve an intra- or extracellular protease(s). IL-38 is mainly reported to exist in two forms—a truncated form and a full-length form. However, which form(s) of IL-38 is naturally present in the human body remains unknown. Interleukin-1 receptor 1 (IL-1R1) and IL-36R are activated by the agonists IL-1 and IL-36, respectively. The heterodimeric receptor complex consisting of IL-1R1 or IL-36R and interleukin-1 receptor accessory protein (IL-1RAcP) results in myeloid differentiation primary response 88 (MyD88) recruitment through the intracellular toll-interleukin 1 receptor (TIR) domain, and the downstream signaling pathways that are activated include (A) the nuclear factor kappa B (NF-κB) pathway and (B) the mitogen-activated protein kinase (MAPK) pathway through extracellular regulated protein kinases (ERK), p38 or c-Jun N-terminal kinase (JNK), which stimulate the activator protein-1 (AP-1). NF-κB and AP-1, then bind DNA and stimulate the production of pro-inflammatory cytokines and chemokines. In contrast, IL-38 acts as an antagonist of IL-1R1 and IL-36R, restraining the recruitment of IL-1RAcP and the binding of agonists, similar to what occurs for IL-1Ra and IL-36Ra. IL-1Ra and IL-36Ra recruit the inhibitory receptor single immunoglobulin IL-1R-related molecule (SIGIRR), which interferes with the TIR domain to exert antagonist properties, it remains unknown whether IL-38 can recruit SIGIRR. In addition, truncated IL-38 antagonizes IL-RAPL1 by recruiting a not-yet-identified inhibitory coreceptor and suppressing the JNK/AP1 pathway, full-length IL-38 activates the pathway.

**Figure 2 biomolecules-09-00345-f002:**
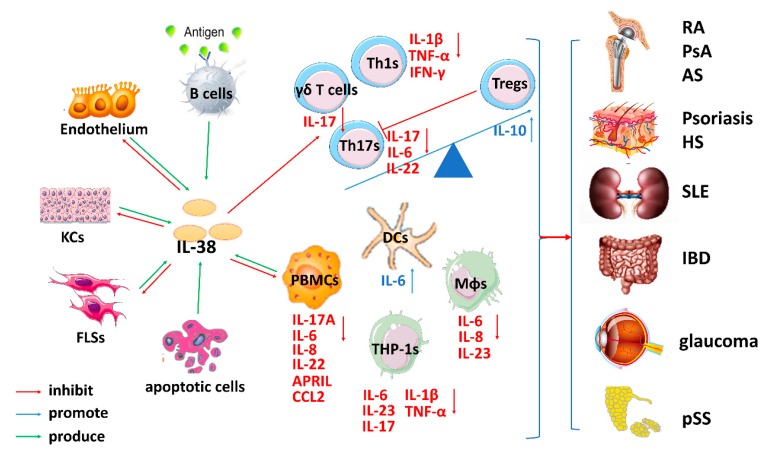
The complicated role of IL-38-mediated immune responses in inflammatory autoimmune diseases. After tissue damage induced by several stimuli, IL-38 is released by B cells, the endothelium, keratinocytes (KCs), fibroblast-like synoviocytes (FLSs), peripheral blood mononuclear cells (PBMCs), and various immune cells, such as dendritic cells (DCs) and macrophages (Mφs). The increased Th1/Th17/γδ T cell numbers and decreased Treg numbers participate in the pathogenic mechanisms of many inflammatory autoimmune diseases. IL-38 inhibits Th1s, Th17s and γδ T cells activated by IL-36 or IL-1 procytokines and increases Treg numbers to decrease TNF-α, IFN-γ, IL-1β, IL-6, IL-22, and IL-17 expression. Additionally, IL-38 exerts bioeffects on PBMCs, human leukemia monocytic cell line (THP-1s), DCs and Mφs to inhibit the cytokines IL-6, IL-8, IL-17, IL-22, and IL-23, which further suppresses the differentiation and expansion of Th17s. Consequently, IL-38 inhibits the development of several inflammatory autoimmune diseases by creating dysfunction in Th1s/Th17s/γδ T cells and promoting Treg expansion.

**Table 1 biomolecules-09-00345-t001:** The subfamily of IL-1 cytokines.

	Cytokine	Family Name	Alternative Names	Receptor	Coreceptor	Property	Processing Required for Optimal Bioactivity
IL-1 subfamily	IL-1α	IL-1F1	IL-1A, IL1	IL-1RI	IL-1RAcp	Pro-inflammatory	No
	IL-1β	IL-1F2	IL-1B	IL-1RI	IL-1RAcp	Pro-inflammatory	Yes
	IL-1Ra	IL-1F3	IL-1RN, ICIL-1RA, IRAP, MGC10430	IL-1RI	NA	IL-1R antagonist	No
	IL-33	IL-1F11	C9orf26, DKFZp586H0523, DVS27, NF-HEV	ST2	IL-1RAcp	Pro-inflammatory, transcription regulating factor	No
IL-18 subfamily	IL-18	IL-1F4	IGIF, IL-1 g, IL1F4	IL-18Rα	IL-18Rβ	Pro-inflammatory	Yes
	IL-37	IL-1F7	FIL1Z, FIL-1ζ, IL-1H4, IL-1RP1	IL-18Rα	SIGIRR	Anti-inflammatory, transcription regulating factor	Yes
IL-36 subfamily	IL-36Ra	IL-1F5	FIL1δ, FIL1D, IL1HY1, IL-1L1, IL-1RP3, IL-36RN, IL-1H3, MGC29840	IL-36R	NA	IL-36R antagonist	Yes
	IL-36α	IL-1F6	FIL1E, IL-1ε, MGC129552, MGC129553	IL-36R	IL-1RAcp	Pro-inflammatory	Yes
	IL-36β	IL-1F8	FIL1η, IL1-ETA, IL-1H2, MGC126880, MGC126882	IL-36R	IL-1RAcp	Pro-inflammatory	Yes
	IL-36γ	IL-1F9	IL-1RP2, IL1E, IL-36G, IL-1H1	IL-36R	IL-1RAcp	Pro-inflammatory	Yes
	IL-38	IL-1F10	IL-1HY2, FKSG75, MGC11983, MGC119832, MGC119833	IL-36R, IL1RAPL1, IL1R1?	Unknown	Anti-inflammatory, proinflammatory?	Yes
NA, not applicable							

**Table 2 biomolecules-09-00345-t002:** IL-38 and its related cytokine expression and function in inflammatory autoimmune diseases.

Inflammatory Autoimmune Diseases	IL-38 and Related Cytokines Levels	Function
Rheumatoid arthritis (RA)	Increased IL-38, IL-36R, IL-36 procytokines, and IL-36Ra levels in the serum and synovial fibroblasts [29,75,77]. Elevated IL-38 levels correlate with IL-1β, TNF-α, IL-6, IL-1Ra, CCL3, CCL4, M-CSF, erythrocyte sedimentation rate and C-reactive protein [29,78]. IL-38, IL-36 procytokines and IL-36Ra correlate with each other [29]. Th17 numbers are elevated but Treg numbers decreased in RA [71].	IL-38 overexpression ameliorates collagen-induced arthritis (CIA) and K/BxN-serum-transfer-induced arthritis (STIA) but not antigen-induced arthritis (AIA) and has no influence on gristle or bone damage [40]. IL-38 neutralization exacerbates RA syndromes [40], and induces greater disease severity in STIA [76]. IL-38 reduces proinflammatory cytokine and chemokine expression in PBMCs, SFs and THP-1 cells [40]. IL-38 decreases Mφ infiltration to reduce expression of Th17 cytokines, TNF-α, chemokine ligand 1 and nuclear factor kappa-B ligand. IL-38 might be a biomarker for RA [78]. Th17 neutralization suppresses the development of CIA and AIA [83,88]. IL-36 signaling neutralization has no effect on RA [63]. IL-1R1 neutralization attenuates progression and severity of CIA [64].
Psoriasis	IL-38 levels are significantly reduced in skin lesions and in circulating of psoriasis [29,44,96] but are increased in pustular psoriasis [56]. Levels of IL-36R, IL-36 procytokines, and IL-36Ra [92,98,99], but not IL-38, are increased in psoriasis [98]. Increased Th1 and Th17 numbers and decreased Treg numbers occur in psoriasis [94]. Anti-TNF-α therapy improves psoriasis and decreases IL-36R, IL-36 procytokine, and IL-36Ra levels [92]. Anti-IL-22 therapy diminishes psoriasis symptoms and reduces IL-36 procytokine expression [51].	IL-38 dampens Th17 responses [94]. IL-38 knockout mice show delayed disease resolution and exacerbated IL-17 inflammation, which can be reversed by adding IL-38 or γδ T cell-receptor antibodies [95]. IL-38 expression is decreased in dedifferentiated KCs [56] and correlates with reduced expression of CK10 [29]. IL-38 correlates with disease severity [97]. IL-38 knockout shows no impact in a mouse model of psoriasis [44]. Th17 numbers and serum IL-17 levels are strongly related to the systemic disease activity of psoriasis and psoriatic arthritis [73,93]. Th17 cytokines can inhibit KC differentiation [107]. IL-1RACP knockout mice show attenuated inflammation by γδ T cells [95]. IL-36R neutralization attenuates psoriasis [100]. IL-36Ra neutralization strongly exacerbates skin inflammation [101], but IL1R1 and IL1RAcP neutralization produces no obvious skin abnormalities [104].
Systemic lupus erythematosus (SLE)	IL-38 and IL-36 procytokine levels are significantly increased in the serum and correlate with disease activity [30,110]. IL-38 expression is significantly decreased in Murphy Roths Large /lpr mice [112]. IL-38 expression decreases after treatment in juvenile-onset SLE patients. [54].	IL-38 treatment attenuates disease severity due to a reduction in PBMC and Th17 numbers and promotes Treg expansion, with no influence for Th1s and Th2s [30,112]. IL-38 neutralization increases proinflammatory cytokine IL-6, CCL2, and APRIL production in PBMCs [30]. IL-38 is associated with complications in the renal and central nervous systems [30].
Inflammatory bowel disease	Increased IL-38, IL-36α/γ and IL-36Ra expression in colonic biopsies from Crohn’s disease patients correlate with IL-1β and IL-17 [29]. IL-38 and IL-36α/γ levels are increased in dextran sulfate sodium-induced colitis [29,114]. IL-36Ra expression in lamina propria mononuclear cells is decreased in ulcerative colitis [29].	IL-36R knockout mice exhibit reduced intestinal inflammation with decreased inflammatory cell infiltration [114,116,117]. IL-36R neutralization elevates Th17 responses while reducing Th1 responses by decreasing IL-2 levels [114].
Other inflammatory autoimmune diseases	IL-38 expression is increased in primary Sjogren’s syndrome (pSS) [57] and hidradenitis suppurativa (HS) patients [119]. Increased IL-36α levels in the serum and salivary glands of pSS patients correlate with disease activity [57]. IL-36 procytokine and IL-36Ra levels are significantly higher in HS [119]. IL-38 and IL-36 procytokine levels are increased in the aqueous humor of glaucoma patients [123].	The IL-36 axis is important in pSS [57]. IL-23/Th17 pathway components are overexpressed in pSS [118] and HS [120]. Th17 numbers as well as IL-17 levels correlate greatly with ankylosing spondylitis activity [73].

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
