# Peer review of "IL-38: A New Player in Inflammatory Autoimmune Disorders"

_biomolecules, 2019, doi:10.3390/biom9080345_

Round 1

Reviewer 1 Report

I appreciate the efforts of authors as they answered my concerns. Over all it has been improved but I feel this manuscript can be written better to make it more readable. It must go through extensive language editing. 

Author Response

Dear Editor: 

the manuscript have undergo extensive language editing by using MDPI ,please see the attachment~

Sincerely

Reviewer 2 Report

In the review article titled “IL-38: a new player in inflammatory autoimmune disorders”, that authors have highlighted the functional role of IL-38 in various signaling pathways. The review also details the role of IL-38 in various autoimmune disorders like psoriasis, SLE and IBD. The review elaborates various aspects of IL-38 in detail with appropriate figures and citations. I recommend this manuscript for publication in this journal.

Author Response

Dear reviewer:

the manuscript have undergo  language editing by using MDPI,please see the attachment~

Sincerely

Round 2

Reviewer 1 Report

I am satisfied with the revision. I would recommend accepting this review in the current format.